# How does handwashing behaviour change in response to a cholera outbreak? A qualitative case study in the Democratic Republic of the Congo

Sian White[1]*, Anna C. Mutula[2], Modeste M. Buroko[2], Thomas Heath[3], François K. Mazimwe[2], Karl Blanchet[4], Val Curtis[1], Robert Dreibelbis[1]

1 Department of Disease Control, London School of Hygiene and Tropical Medicine, London, United Kingdom, 2 Independent Consultant, Goma, Democratic Republic of the Congo, 3 Action Contre la Faim, Paris, France, 4 Geneva Centre of Humanitarian Studies, Université de Genève, Geneva, Switzerland

* Sian.white@lshtm.ac.uk

**Data Availability Statement:** The supporting dataset for this work is available via Figshare at the following link at the following citation: White, Sian

## Abstract

### Background

Handwashing with soap has the potential to curb cholera transmission. This research explores how populations experienced and responded to the 2017 cholera outbreak in the Democratic Republic of the Congo and how this affected their handwashing behaviour.

### Methods

Cholera cases were identified through local cholera treatment centre records. Comparison individuals were recruited from the same neighbourhoods by identifying households with no recent confirmed or suspected cholera cases. Multiple qualitative methods were employed to understand hand hygiene practices and their determinants, including unstructured observations, interviews and focus group discussions. The data collection tools and analysis were informed by the Behaviour Centred Design Framework. Comparisons were made between the experiences and practices of people from case households and participants from comparison households.

### Results

Cholera was well understood by the population and viewed as a persistent and common health challenge. Handwashing with soap was generally observed to be rare during the outbreak despite self-reported increases in behaviour. Across case and comparison groups, individuals were unable to prioritise handwashing due to competing food-scarcity and livelihood challenges and there was little in the physical or social environments to cue handwashing or make it a convenient, rewarding or desirable to practice. The ability of people from case households to practice handwashing was further constrained by their exposure to cholera which in addition to illness, caused profound non-health impacts to household income, productivity, social status, and their sense of control.

(2022): Interviews and group discussions with crisis and outbreak affected populations in the Democratic Republic of the Congo on handwashing determinants. figshare. Dataset. https://doi.org/10.6084/m9.figshare.19469270.v1 Please note that this includes transcripts in English of all interviews and focus group discussions which have been redacted to remove any identifiable information. Interested researchers may also contact the corresponding author or the Research Governance and Integrity Office at LSHTM: RGIO@lshtm.ac.uk to access additional data.

**Funding:** SW and TH recieved the funding from the United States Agency for International Development's Bureau of Humanitarian Assistance (Grant number: AID-OFDA-G-16-00270). Donor website: https://www.usaid.gov/who-we-are/organization/bureaus/bureau-humanitarian-assistance The funders had no role in study design, data collection and analysis, decision to publish, or preparation of the manuscript.

**Competing interests:** The authors have declared that no competing interests exist.

## Conclusions

Even though cholera outbreaks can cause disruptions to many determinants of behaviour, these shifts do not automatically facilitate an increase in preventative behaviours like handwashing with soap. Hygiene programmes targeting outbreaks within complex crises could be strengthened by acknowledging the emic experiences of the disease and adopting sustainable solutions which build upon local disease coping mechanisms.

## Introduction

For centuries cholera has been a marker of social inequalities, affecting the most vulnerable members of society and commonly occurring amid social and economic upheaval or disaster [1]. Cholera cases remain underreported but it is estimated that there are 2.9 million cases and 91,000 deaths annually due to the disease [1, 2]. In 2017 major outbreaks occurred in Yemen, the Democratic Republic of the Congo (DRC), Nigeria, Somalia and South Sudan leading to the highest global numbers of cases in history [3].

In the past, cholera was viewed as a waterborne disease, with environment-to-human transmission of *vibrio cholera* believed to be responsible for the majority of transmission [4]. However, recent spatiotemporal analyses of cholera outbreaks have demonstrated how cases cluster among close contacts [5–8]. This human-to-human transmission is heightened in dense living environments, and where water access is limited or intermittent, causing hygiene to be compromised [8].

Handwashing with soap is frequently recommended by international response agencies as a key household-level cholera prevention behaviour [9]. A recent meta-analysis of case control studies conducted during cholera outbreaks found that self-reported good hygiene practices and the availability of handwashing materials had the highest protective affect of any of the water, sanitation and hygiene (WASH) factors assessed [10]. Another broader review of cholera risk factors also found that handwashing had smaller but still protective effect against symptomatic cholera [8]. The authors acknowledged that the included studies used inconsistent measures of self-reported behaviour, likely to result in overestimates of actual handwashing behaviour [11].

Another review assessing the impact of water, sanitation and hygiene (WASH) interventions on cholera control found that handwashing promotion programmes during outbreaks generally had a positive affect but were limited by the behavioural and health outcomes they used (e.g. self-reported symptoms rather than laboratory confirmation). A more recent study in Bangladesh was able to overcome measurement limitations, and demonstrated that case-targeted interventions to promote handwashing and water treatment were successful in increasing behaviour which consequently reduced secondary transmission to household contacts by almost half [12]. However the majority of handwashing interventions during cholera outbreaks continue to focus only on health education [13]. This is problematic because knowledge of the health benefits of handwashing is unlikely to be sufficient to realise sustained behaviour change [14, 15]. During outbreaks there is a tendency for both researchers and practitioners to overemphasise the effect of cognitive determinants such as health knowledge, risk perception and fear, rather than taking a more holistic view of the determinants that could influence handwashing behaviour [15]. Outbreaks may also cause theory-informed processes for designing behaviour change programmes to be compromised due to the perceived need to act right away, rather than consult and learn from populations [16].

There have been calls for more qualitative research into hygiene behaviour during cholera outbreaks and new, scalable approaches to doing community engagement to support preventative behaviours [1, 17, 18]. This research responds to these calls and draws on anthropology and behavioural science to understand how individuals and communities experience and respond to outbreaks and whether this affects their handwashing behaviour. We explore both the consequences of cholera on people's lives and the determinants of handwashing behaviour during the 2017 cholera outbreak in the eastern part of DRC.

## Methods

### Study site

This study took place in South Kivu in the eastern part of the DRC at the height of the 2017 cholera outbreak (October and November). The region experiences both endemic and epidemic cholera and in 2017 the outbreak was the largest in recent decades with >53,000 reported cases and 1,145 deaths [19]. The research took place in a town on the shores of Lake Kivu (known to be an environmental reservoir for cholera) [20, 21], which is home to about 200,000 people. The region was purposively selected because it was described as a cholera 'hot spot'[19]. It also hosts a large number of internally displaced persons (IDPs) who have fled armed conflict in neighbouring villages. At the time of this research, the government provided IDPs with a small plot of land (~3m2) within one of two informal camps in the town. IDPs were responsible for constructing their own makeshift shelters from tarps and branches. In addition to these camp-like settings, some IDPs rented homes from permanent residents. Host community members typically lived in brick or compacted mud houses with corrugated iron roofs. Both IDPs and host community members typically worked in agriculture or as small-scale market vendors, although IDPs would typically earn less than host community members on a daily basis. Water and sanitation access in the region was poor. Pit latrines were common but often in poor condition and shared by many households, particularly in the informal camps. Water was considered scarce and was intermittent. IDPs and host communities had access to the same water sources which included tap stands and boreholes or water collection from rivers and lakes.

Multiple non-government organisations (NGOs) had worked in the region as part of sporadic emergency response initiatives oriented towards health and WASH. At the time of the research a temporary Cholera Treatment Centre (CTC) had been established by an NGO and was providing free care for cholera cases. Handwashing promotion was widespread and predominantly consisted of health education delivered by volunteer *relais communautaires* (health volunteers from the community) who were trained by NGOs. Health awareness sessions focuses on cholera transmission and prevention behaviours. Exposure to hygiene promotion was similar among IDPs and host communities, with the exception that those living further outside of the town in were exposed less frequently.

### Research framework

This research used unstructured observations, in-depth interviews (IDIs) and focus group discussions (FGDs) to explore a range of behavioural determinants. We used the Behaviour Centred Design (BCD) framework [22] to develop a list of determinant categories and to refine appropriate methods for exploring each. BCD draws on evolutionary and environmental psychology to define critical domains of behaviour including cognitive processes, individual characteristics, the settings where behaviours take place and the broader physical, social and contextual environment. S1 Table defines the 16 BCD determinant categories that were assessed within this research in relation to handwashing. It indicates which methods were used

to explore them and how these methods were developed. All of the methods aimed to identify contextual associational relationships within these broader determinant categories. The methodology adopted in this research replicated a process used in Iraq to understand how behaviour was affected during post-conflict displacement [23].

## Sampling

For the observation and IDIs participants were selected purposively based on their exposure to cholera. To do this we worked with health staff to identify cholera cases registered within the last three months. Using this sampling frame, we purposively selected for a diversity in age, gender, geography (rural or urban) type of residence (residing in a camp or residing in the community). Once case households had agreed to participate in the research, we also approached other households in the nearby vicinity to be part of the research. We sampled these 'comparison households' against the same criteria for diversity. Some households participated in more than one method. Sampling continued until a degree of saturation was met for each method. FGD participants were sampled purposively to be similar in terms of gender, geographical regions and type of residence.

## Data collection methods

**Unstructured observations.**    Unstructured observations were designed to provide contextual detail about handwashing. Observations took place in eight case households and eight comparison households. Observations were for 2–3 hours, typically beginning at 6am and finishing when the participant had to depart for work. Observers wrote down all actions that that were done by all household members and the time actions took place. Observers paid attention to 'critical times' for handwashing which were defined as handwashing after using the toilet, or cleaning a child's bottom, and before preparing food, eating food or feeding a child. Observers noted whether hands were washed at these critical times and whether soap was used.

**In-depth interviews.**    A total of 51 IDIs were completed, involving 24 people from case households and 27 people from comparison households. Seven of the participants who took part in the observation also took part in the IDIs. For case households, we selected the individual who had cholera if they were over 18 and well enough to participate. Alternatively, the person primarily responsible for caring for the case was selected. Participants of a similar age and gender were then invited to participate from neighbouring comparison households. A total of eight participatory activities were used within the IDIs to explore perceived hygiene challenges, the enabling environment, water use, roles, capabilities, routines, norms, social networks, and broader contextual determinants. See S2 Table for details on these activities.

**Focus group discussions.**    43 people participated in the FGDs. Four FGDs were conducted with women, two of which comprised women residing in the IDP camps and two which were with host community members. Three FGDs were conducted with men, two of which were with IDPs and one with host community members. Six participatory activities were included in FGDs to explore the prioritisation of hygiene challenges, perceived risk of cholera, attitudes towards people who had cholera, preferences related to infrastructure and soap, and motivations of behaviour. See S3 Table for descriptions of these activities.

**Data collection.**    The data was collected by four of the authors (SW, ACM, MMB, FKM). We were a team with mixed cultural backgrounds (British and Congolese). Two days of classroom-based training was provided on the research rationale and the methods. We then piloted the methods in a similar setting and adapted the tools as necessary. All IDIs and FGDs were conducted in Congolese Swahili and audio recorded. Observation notes were taken by hand.

At the end of each day of data collection we reflected on our findings and captured this through written field notes [24].

**Data management and analysis.** Preliminary data analysis was done concurrently with data collection. This allowed us make theoretical and methodological notes [25] and decide when we had reached a point of saturation. All audio recordings from IDIs and FGDs were transcribed and translated. Methods with semi-quantitative data such as ranked or scaled information were summarised in spreadsheets. Visual data such as drawings, photos and videos were descriptively summarised. All data and the field notes were imported into NVivo 12 software. The data analysis followed the process outlined by Braun and Clarke [26]. Data were classified according to whether the participant was from a case household or a comparison household. An initial top-down coding framework was applied based on the determinants of the BCD checklist. A second phase of coding was then conducted based on emergent themes. Coding was conducted by the first author and then theme summaries were validated by ACM, MMB, FKM and TH.

**Ethics and consent.** Informed written consent was obtained from each participant. The research was approved by the ethics committees at the London School of Hygiene and Tropical Medicine (Submission ID: 13545) and the School of Public Health at the University of Kinshasa (Submission ID: ESP/CE/038/2017). Permission to undertake this work was given by the Departments of Health in North and South Kivu. Organisations working in the area were informed of our work and preliminary findings were shared immediately after data collection to enable utilisation within programmatic work. Further detail on research permissions and stakeholder engagement is provided in an Inclusivity Questionnaire in S4 Table.

## Results

### Participant characteristics

In total 104 people took part in this research with 40% of these coming from households with one or more cholera cases in the last 3 months. A higher proportion of women were included in the sample, this purposive selection reflected the fact that women in this region were more involved with hygiene-related tasks and caregiving. Almost half of the participants were illiterate and average family size was 6.5 people. The linguistic diversity of participants and the high levels of internal displacement and people returning post-displacement (62% experiencing displacement) are reflective of decades of conflict in this region. Table 1 summarises the socio-demographic characteristics of the sample.

**Handwashing behaviour.** Handwashing with soap (HWWS) and hand rinsing were uncommon at critical times (HWWS = 1%, hand rinsing = 11%) and there was no discernible difference between behaviour in case households and comparison households or between IDPs and host community hosueholds. When handwashing with soap did occur, it was typically performed following dirty household cleaning tasks (such as sweeping or cleaning the toilet). Hand rinsing was commonly practiced before eating and feeding children. While people knew that ash could be used as an alternative to soap, this practice was not seen during any of the observations.

**Determinants of handwashing behaviour.** We identified a range of context specific determinants of handwashing behaviour across the 16 BCD categories (See S5 Table for full list of determinants and their association with handwashing behaviour). Some of the determinants assessed appeared to have no impact on handwashing behaviour including ethnicity, religion, education level, sociality, access to ash, descriptive norms, knowledge about key handwashing moments, and the motives of comfort and affiliation. Below we describe determinants that had a reported positive or negative influence on behaviour.

**Table 1. Socio-demographic characteristics of all participants across the two camps and two villages.**

| Socio-Demographic characteristics | Total N = 104 | % |
|---|---|---|
| **Sex** | | |
| Male | 37 | 36% |
| Female | 72 | 69% |
| **Place of residence** | | |
| Camp | 22 | 21% |
| Community | 82 | 79% |
| **Location** | | |
| Rural | 34 | 33% |
| Urban | 70 | 67% |
| **Literacy** | | |
| Not literate | 49 | 47% |
| Literate / Some literacy | 55 | 53% |
| **Cultural diversity** | | |
| Number of local languages spoken across participants | 8 | |
| **Household Size** | | |
| Range | 1–13 people | |
| Average | 6.5 people | |
| **Displacement status** | | |
| Internally Displaced | 51 | 49% |
| Returnee (a person who has been displaced in the last 5 years and has returned to their home) | 13 | 13% |
| Host community | 40 | 38% |
| **Duration of displacement for IDPs** | | |
| Range | 1 month to 20 years | |
| Average | 4.5 years | |
| **Cholera exposure** | | |
| Participants who had one or more cholera cases in their household in the last 3 months | 42 | 40% |
| Households with no recent direct exposure to cholera | 62 | 60% |
| **Duration since cholera cases were discharged** | | |
| Range | 1–90 days | |
| Average | 17 days | |

*Knowledge*. All participants were familiar with cholera, its symptoms (e.g. mentioning cholera-specific symptoms like 'rice water' stools), and recommended prevention behaviours. However, cholera was often used as a catch-all term to describe a range of diarrhoeal diseases. Handwashing knowledge was high with all participants able to list critical moments for handwashing and explain how handwashing can interrupt disease transmission. Participants attributed their familiarity with cholera and preventative behaviours such as handwashing to frequent exposure to hygiene promotion activities.

*Physical environment and behavioural settings*. Across the various settings within study site, there was little in the physical environment to enable or cue handwashing at key times. In discussions with participants, they would often differentiate between handwashing being easy to do as a behaviour, but difficult for them to practice because they lacked the products (soap) and infrastructure (water and handwashing facilities) which could facilitate it. Most research participants were agricultural labourers and spent the majority of their days outside the home. During this time people typically had no access to soap or water, preventing handwashing

from taking place. Within the camp settings, NGOs had constructed simple bucket-style hand-washing facilities and in some of the rural settings tippy-taps had been promoted. However almost all of these were non-functional at the time of our research. Participants also admitted that even when these facilities were functional, water, soap and ash were not readily available at the stations. In FGDs participants agreed it was important to have "somewhere special" for handwashing. They felt handwashing stations acted as a reminder to wash hands at key times and helped to inculcate good habits in children. However, the basic handwashing facilities promoted by NGOs were seen as being "poor designs, for poor people". They tended to break easily and therefore failed to have a lasting impact on behaviour. Participants felt handwashing facilities should symbolise hygiene, rather than just facilitating handwashing:

> *"It is essential to have a beautiful, an attractive hand washing facility so one is at ease when washing hands. . . the hand washing facility has to be always kept clean so that it does not disgust, and a facility has to be respected by the whole family and everyone."* (Male FGD participant)

Almost all participants reported that water scarcity was a major barrier to hand hygiene and a source of stress within their lives. In general, participants did not think handwashing consumed much water. However, water access and usage was carefully calculated and prioritised for other household tasks like bathing, cooking, laundry and dishes. While water was often sectioned out for different purposes within the home, but no families felt they could easily put water aside for handwashing. Observations indicated people used a range of water for handwashing including washing their hands directly in nearby lakes and rivers and re-using grey water (e.g. water from dishwashing or laundry).

Soap was a valued and scarce commodity. Overall, 62% of households had no soap of any kind available at the time of our visit. Among those who did have soap, it was typically kept on a high shelf in the bedroom and therefore not conveniently available for handwashing. No one reported buying soap just for the purpose of handwashing. Rather, soap was typically purchased when laundry needed to be done and then a small leftover section may be used for bathing and handwashing. While laundry soap was affordable it was not desirable for handwashing:

> *"You know, this [laundry soap] is not a soap we want to use because it can damage hands, but we just use it because of poverty."* (Female FGD participant)

As with water, soap use within the household involved conscious trade-offs and decision-making between family members:

> *"Getting soap is not easy. . .In a house of nine children, you understand that a piece of soap will not be prioritized for hand washing. If we manage to afford the piece of soap for 100 Congolese Francs only once a week. . . How, and on what, can you use just a small piece of soap? Will you use for laundry? For bathing? Or for hand washing? Things become complicated."* (Female FGD participant)

Participants explained that it was relatively common to ask neighbours for soap and water if needed but said people would laugh at you if you asked for these items for handwashing.

The social environment—norms, routines and social influence

Daily routines were unpredictable for most of the participants, with individuals searching for employment in the fields of others on a day-to-day basis. Daily routines were further

complicated by intermittent water supplies and individuals could spend several hours per day searching for water they considered to be safe. Combined, these factors created time and financial pressure. The irregularity of routines, and the daily stressors that accompanied this, decreased the likelihood of handwashing habits forming. This was because there were few routine sequences of behaviour to cue handwashing and there was limited ability to make plans related to handwashing (e.g. to budget to have enough soap in the house).

Handwashing was seen as socially desirable and an injunctive norm. Hand hygiene norms were heightened by the outbreak, with participants estimating 65% of their community had increased their frequency of handwashing due to cholera concerns. Despite this, social sanctions or judgment related to not washing hands was low:

> *"I cannot judge people around me for not being clean because sometimes I am not clean too. . . I think only about 30% of people would judge me negatively [if they saw me not washing my hands] because people are not focused on hand washing behaviour; they can take it as normal that people sometimes forget to wash their hands."* (Female IDI participant from a comparison household)

Participants explained that people easily forgave each other for not washing hands because of their difficult circumstances. Social support for handwashing was limited. Participants felt handwashing could be facilitated by family members reminding each other to wash hands at key times, but this rarely happened in practice. The importance of handwashing behaviour was consistently reinforced by NGOs and while some people felt the repetition of handwashing messages helped to remind people, others found it frustrating that these organisations were unable to realise changes to their broader circumstances which would allow them to practice handwashing more regularly.

Motives

In FGDs participants were asked about which motives were associated with handwashing. Motives of love, attractiveness and status (e.g. wealth, education and social respect) were thought to be strongly associated with handwashing:

> *"No one can fall in love with someone with dirty hands!"* (Female FGD participant)

> *"An attractive person is likely to remember to wash hands because she is used to looking nice, so, her hands have to look nice as well."* (Female FGD participant)

> *"Highly educated people are always clean because they do work with white papers and with clean things, and so they also have to have clean hands. And again, these educated people teach others; they cannot go in front, teaching others when they are unclean. They have to have clean hands so that people consider and respect them."* (Male FGD participant)

Handwashing was less strongly associated with nurture and was not seen to be associated with affiliation (fitting in with a group). Participants explained that even 'good parents' are unable to mind their children and encourage handwashing behaviour because everyone has to work long hours outside the home. Others explained that relationships were typically built on shared interests and needs and so handwashing didn't necessarily help a person to fit in. Participants in IDIs and FGDs agreed that if a person was hungry, poor or upset they would be unable to prioritise handwashing:

> *"Hygiene and good nutrition work together. Cleanliness cannot be visible in a house where food is absent. I can't think about handwashing when I am so hungry. Another thing is the*

*kind of life I live, since I am IDP, it has brought me to trauma or psychological problems. Hygiene has become difficult because of the many thoughts crossing my mind like: How my children are going to eat? How am I going to get money? It's difficult.*" (Female IDI participant from a comparison household)

Hunger emerged as a particularly prominent barrier to hand hygiene in this setting. At critical moments for handwashing, such as preparing or eating food, people's hunger would override all else, causing people to forget handwashing. Hunger also caused prevented people from making plans that could facilitate handwashing. For example, participants explained that their limited daily earnings are entirely consumed by purchasing food:

*"If I do not go to work in the farms all day, we shall not eat. . . But if you are only able to earn 2500 Congolese Francs, then all of this money will have to go on food. We ask ourselves whether to buy soap. . . but it is difficult to choose this rather than prioritizing food for our children."* (Female FGD participant)

*Demographic characteristics* Certain personal characteristics influenced handwashing behaviour. Older adults and men who lived alone often were unable to collect sufficient water to meet their needs and therefore reported making handwashing compromises. Their reduced water availability was due to accessibility or cultural barriers associated with gender norms (i.e. generally it was only women who collected water). IDPs typically faced more challenging living conditions than host community members and often described feeling that they were living "like animal"' or "living a life that was not our own". Consequently, many IDP participants reported that the 'problem of handwashing' was new for them and if their old lives could be restored their behaviour would also improve:

*"I can tell you that I had a good life, I was a rich person. . . but my life changed with displacement. . .the situation changed to bad and today I am as you see me. take me back to my previous life and you will see my feelings and emotions will change and then I will have a high chance of washing his hands with soap."* (Male FGD participant)

**Variations in experiences of cholera and behavioural determinants between participants from case and comparison households.** The determinants of handwashing behaviour among case households were consistent with those in comparison households across the determinant categories described above. However, there were substantial differences between the two groups on cholera-related risk perceptions and the perceived or actual consequences of cholera on people's livelihoods and routines.

*Participants from comparison households.* Cholera was the main health concern of almost all participants, but among those who had not had a cholera case in their household, it was seen as a common, chronic challenge that was inseparable from other adversities they faced:

*"Cholera is a major health problem here . . . because we work hard, earn less, eat less and rest less. As consequence, we lose weight and look pale and have poor nutrition. . . Then it is easy for the cholera disease to attack people."* (Female FGD participant)

*"It has been a long time since we do not have drinking water in this area and that is the reason why cholera disease attacked people. . . if you have bad food, unhealthy water and your hands are dirty then of course you will suffer from cholera."* (Male FGD participant)

The high number of cases in the 2017 outbreak did heighten perceptions of risk and participants from comparison households reported realising cholera was serious within recent months. Participants from comparison households thought cholera generally affected children or people who were already 'sickly', 'unclean' or 'pale'. Host community members thought that cholera was primarily a problem that affected IDPs. In terms of the social and economic impacts of cholera, participants from comparison households said they feared people who had cholera and would avoid them while infectious. Those without direct personal experiences of cholera thought its impacts on the lives of cases or their families would be relatively temporary, given that disease was seen to be easy to treat and participants felt that people tended to recover quickly. Participants were divided about whether getting cholera would have a longer-term impact on a person's reputation:

*"Someone who has had it [cholera] would not like people to know that he had it because cholera disease really affects a person's dignity, nobody can wish to get it. It leaves you with no reputation at all."* (Male FGD participant)

*"Everyone understands. . .I mean most people know someone who has experienced cholera at some point. . . So the person who gets infected, he can still recover and get back to his normalcy."* (Male FGD participant)

In many ways local explanatory disease models were aligned with 'Western' biomedical messaging about the cholera (presumably because of the history of health promotion in the region). However, misperceptions about cholera persisted within the community despite familiarity with the disease. Several participants said others in their community do not take cholera seriously because they believe "black people don't die of germs". Similarly, some people believed that a certain degree of exposure to "dirtiness" helps to protect you because being too clean may leave you vulnerable to infection. Others felt the continued presence of cholera in Eastern DRC played to the interests of humanitarian organisations:

*"We know that when [organisations] come just sensitizing about hygiene and cholera, they are paid on this—it's no use coming all the time disturbing us. They come for their own interest."* (Male IDI participant from a comparison household)

Over decades people had realised that most humanitarian aid was provided during cholera outbreaks. These were short term projects which subsided as cholera cases decreased. People had also learned how to make the most of a system that didn't always appear to have their ongoing interests at heart. For example, on several occasions during our research people from the community who were not participants, approached us with lists of 'cholera cases' who needed help.

*Participants from case households.* Participants from case households reported that they were easily able to recognise cholera symptoms. All of these households explained that cholera had affected them suddenly and unexpectedly. Some participants said this in a literal sense, reflecting that people commonly went from feeling healthy to suddenly experiencing vomiting and diarrhoea which lead them to become so weak that they were unable to do anything within hours. More commonly, when participants said cholera was "unexpected" they were expressing that they struggled comprehend how the disease had been able to launch a "surprise attack" on a family like them. Participants therefore attributed their illness to a one-off "mistake" in their behaviour.

When the disease initially "attacks", family members reported being worried. Patients described feeling "empty", "not of this world" and "seeing only death before them". When patients returned home from the CTC, their cholera experience was not over. Cholera cases described feeling "stuck" and "destabilized" in multiple aspects of their life. Patients were unable to do agricultural work or household tasks for about a month because they felt "weak like paper". Given the majority of households in this region survived day-to-day, earning less than 3000 Congolese Francs per day (1.5 USD), this inability to work rapidly put families in a state of economic crisis:

*"I cannot say that my economy decreased—it was totally blocked!"* (Female IDI participant who had cholera)

*"A huge economic impact was observed. . .during that period of our child's sickness, things really got harder. Although we generally eat badly, that particular time my husband was staying with the child at the hospital, we then ate more badly than usual because of the little amount of money—everything went really bad."* (Female IDI participant who had a child with cholera)

People also reported that cholera affected their roles, responsibilities, and sense of self. Participants felt cholera caused their attractiveness to "fade" and that they now "hated their outlook" due to the amount of weight they had lost. Parents who had been cholera cases felt worried that in the months following their discharge they had become unable to care and provide for their children:

*"I became like a baby. . .I had to wait for somebody else to take care of me, like the neighbours, it is like I have lost my role of mother to all these children."* (Female IDI participant who had cholera)

*"Suffering from cholera reduced considerably my responsibility as a father, I could not feel respected, and I could not feel myself as a father of the family because I was half-dead."* (Male IDI participant who had cholera)

Another woman explained that she relied on her neighbour to breastfeed her new born baby for several months while she was sick with cholera. Children and older members of the household often had to stand in for parents to do the household chores. Relationships with neighbours and friends also changed. Some participants explained neighbours were integral in helping them through their recovery and that they often gave them food, money and water (although it was expected the household would find a way to pay this back):

*"I am still feeling weak and have no money to buy food because I am not working, so some neighbours give me food to eat, but you understand how difficult it is to depend on someone else's kitchen."* (Female IDI participant who had cholera)

Others explained that neighbours and friends stayed away following their illness and that they felt isolated and stigmatised:

*"The relationship with neighbours does change because of gossip, they start saying that it is because of your uncleanness. . .they end up avoiding you"* (Man IDI participant who's elderly brother had cholera)

*"I lived an isolated life during those hard periods of cholera cases."* (Female IDI participant who had 3 children who had cholera)

However, participants from case households reported being more motivated to wash their hands, explaining this was because they recognised their vulnerability to disease. They also used handwashing as a way of countering any misperceptions from neighbours about their cleanliness. The latter concern seemed to prompt a range of demonstrative action around handwashing. For example, several participants said they have actively encouraged others to practice handwashing:

*"Though we got infected, we cannot feel discouraged from doing hygiene practices. . .we even try to improve it and we are telling our neighbours that they should keep on practicing hygienic behaviour because that is the only way to prevent cholera. . .And other people around here, when they see how clean you are, even if you got sick with cholera, they can decide to take you as their good example of cleanliness"* (Male IDI participant who had 2 children with cholera)

One participant built a dedicated place for handwashing near the toilet after his daughter was admitted to hospital with cholera:

*"I realise that fighting this cholera disease is very serious these days and so I thought that once we are practicing hand washing, we will be able to prevent our family from this disease. . .So I decided I had to make the place for handwashing, a place that would be respected. . . Now the neighbours are just appreciating [the facility] and I am telling them to do the same as I did, but whether they agree or not, I will never give up with the practice."* (Male IDI participant whose daughter had cholera)

However, households with cholera cases also felt their circumstances following their exposure to the disease made it more challenging to practice handwashing. This was due to their reduced physical health, their inability to collect sufficient water (most households reported being able to access half as much water in the period post discharge as compared to their normal circumstances), increased hunger and malnutrition (due to loss of income), and difficulties affording other basic daily necessities, such as soap. A minority of participants were given a small bar of soap and six water purification tables upon discharge from the CTC. Participants who received these distributions, reported trying to use these sparingly to make them last as long as possible.

## Discussion

Our research found that in the Eastern region of DRC, cholera is generally conceptualised as a persistent and commonplace health challenge but also one that is easily treatable. Frequent hygiene promotion sessions in this region have led to high levels of knowledge about the health impacts of cholera, its symptoms and recommended preventative behaviours. However, handwashing with soap was observed to be rare in this setting. By using theory-driven qualitative methods we were able to identify that this was because the psychological, social, and environmental behavioural determinants affecting handwashing in this context combined to limit the ability of individuals to improve their handwashing behaviour. Major barriers to handwashing related to the physical environment or behavioural setting included the absence of handwashing facilities, water scarcity, the unaffordability of soap, the small make-shift houses where displaced populations lived, the use of shared sanitation facilities, and the extended periods

people spent working outside of the home. Handwashing behaviour was also hampered by broader experiences of living in poverty and within in a dynamic conflict-prone region with high rates of displacement and livelihood fragility. This was because handwashing was often deprioritised because of hunger, mental health challenges, the unpredictability of routines, and the lack of social support and sanctions around handwashing. The experiences of participants from case households indicated that in complex crises, cholera can have profound non-health impacts on a household's income, productivity, social status, and sense of control–factors which in turn create additional barriers to handwashing.

Despite low rates of actual handwashing practice, our research participants reported hand-washing had increased as concerns about cholera were heightened. Many participants felt these changes in behaviour might be sustained beyond the outbreak. Prior literature has indicated self-reported handwashing behaviour tends to increase during outbreaks [15, 27–31], however, studies which use observational measures of behaviour show much lower rates of practice even during outbreaks [32]. Such findings act as a reminder that research exploring handwashing behaviour should prioritise including observational methods to gauge actual practice given that self-reported behaviour is commonly affected by social desirability bias, and that this bias may be heightened in outbreaks [33]. However, it may also be indicative how behaviour may fluctuate over the course of outbreaks. For example, initial gains in the frequency of handwashing behaviour at the onset of an outbreak seem to decline or vary over time as fear associated with the disease subsides or the disease is normalised [27, 34, 35]. One handwashing study published during the COVID-19 pandemic suggested that such patterns in behaviour may be explained through Terror Management Theory (TMT) [35]. This theory suggests that when the threat to our mortality from a disease is made more salient, we are more likely to adopt health behaviours, like handwashing, that can remove this threat from our focal attention [35–37]. This theory also explains that when the disease threat is no longer the focus of our attention, protective behaviours may start to decline. In our study participants were aware of the proximal threat of cholera in their region but adopted other psychological defences (such as perceiving others to be at greater risk than them and believing "black people don't die of germs") which avoided the threat and made this reality easier to cope with on a day-to-day basis. TMT might also explain why case households were more driven to take demonstrative action around handwashing following their recent brush with death. In contrast, comparison households in our study site were pre-occupied by more salient threats to their mortality such as hunger and conflict. As such their daily behaviour was geared to the reduction of these threats rather than cholera prevention behaviours. There are few studies which explore how stress or external threats may affect the prioritisation of handwashing behaviour, however, consistent with our results, one study among health care workers in a high-income setting indicated that stress, cognitive load and threats to mortality that appear more urgent or proximal, may impair a person's ability to practice handwashing [38]. Our findings challenge the common belief that if people understand the benefits of handwashing they will act 'rationally' during an outbreak and wash their hands more frequently to protect themselves and others [16, 39].

In our study, households with cholera cases experienced the disease as an exogenous shock to their already vulnerable state which plunged their household into a state of acute socio-economic crisis. This household-level crisis was characterised by a sudden but extended loss of income, increased hunger, isolation from social support systems, feeling unable to provide for family members, and feeling that their exposure to cholera may tarnish their social standing in the long term. These lived experiences of cholera are consistent with existing, but limited, literature from other settings where cholera outbreaks occur during complex crises or within fragile states and among populations with high levels of poverty [40, 41]. Our study found that

exposure to cholera decreased the household's access to food and made it hard to prioritise handwashing due to a reduced ability to access water purchase soap in the wake of their illness. This presents a critical challenge for cholera control given that hand hygiene is likely to be key to interrupting transmission during the 10 days when cases are hyper-infective following infection [42] and *v. cholerae* continues to be shed in their faeces. There is also some evidence that pre-existing and continued malnutrition during this period may prolong shedding [6]. Our findings support the likely effectiveness of targeted WASH interventions distributions of hygiene kits [12, 43] and suggests that these could be complemented by the distribution of food items in some settings.

Prior research in this region of DRC has highlighted that hygiene programming may be met with reduced acceptability if it is inadequately resourced, poorly contextualised, fails to acknowledge other priorities of the population, or does not address social and environmental factors that may constrain behaviour [44, 45]. Broader research has also indicated that during complex crises, experiences and responses to cholera outbreaks are associated with, and amplified by, structural and social vulnerabilities such as extreme poverty, conflict and displacement [46–48]. Our findings are consistent with this body of research and indicate that hygiene programming aimed at mitigating cholera transmission amid complex crises is likely to be more effective if it is integrated into longer-term initiatives that focus on these larger vulnerabilities, such as food security, livelihoods and psychosocial support initiatives. In contexts where cholera is endemic, handwashing programmers must move beyond health-education and work with communities to build enabling environments through investment in handwashing facilities and reliable water supply systems, and supportive social structures. Participants in our study highlighted the importance of conveniently located, desirable and durable facilities in cueing behaviour at key times and this is supported by broader literature [15]. Our research also identified examples of adaptive coping strategies utilised by the population to facilitate handwashing behaviour, reduce vulnerability, and increase their sense of control over the unpredictability of their circumstances. Coping strategies included the use of surface water or grey water for handwashing, the use of ash when soap was unavailable, the pooling or water and soap resources within compounds, the careful calculation of water and soap use to facilitate all necessary household tasks and encouraging neighbours to remind all children within a compound about handwashing. While these actions were taken by a minority of households in our study, they could easily be shared and adopted by others by utilising a positive deviance approach [49]. Research in previous outbreaks has highlighted the importance of understanding whether local coping mechanisms are aligned with, juxtaposed to, or are able to fill gaps in government and organisation-led disease prevention strategies [50–52]. Experiences during prior outbreaks has also emphasised that an overreliance on biomedical explanations of disease can be met with resistance from populations [16, 53, 54]. If health promotion fails to acknowledge emic perspectives and experiences it has the tendency to isolate the disease from its human host and the social experiences that facilitate transmission [55]. Our findings suggest that handwashing programmes should aim to change the public health discourse around cholera-related risk by focusing on local constructions of disease, the experiences of populations, and by communicating the non-health impacts of the disease. This may allow populations to adjust their decision-making and coping mechanisms towards prioritising behaviours like handwashing—particularly if it is seen to have health, social and economic benefits in the long term. Lastly, our research found that case households were more motivated to practice handwashing after their exposure and were better able to act upon their behavioural intentions to encourage the behaviour in others and create an enabling physical environment for handwashing. Humanitarians could build upon this by inviting cholera cases to share their experiences with others in the community. There is some evidence that this may be an effective way to

motivate health behaviour, challenge misperceptions around diseases and to heighten perceived vulnerability in a way that is more sustainable than focusing on fear alone [56].

## Limitations

Our research was primarily interested in exploring how the determinants of handwashing behaviour were affected by a cholera outbreak. While observed and self-reported behaviours are described qualitatively in this study, the methods were not designed to be representative and therefore this data could usefully be complemented by further research which measures actual behaviour before, during and after outbreaks in regions that are prone to them.

Where possible we used participatory activities that have been used in prior research however some new approaches were developed to explore motives, water prioritisation and experiences of conflict. Replication of these methods would be useful to demonstrate their validity and reliability.

Our sampling was guided by case lists from the CTC, however in this region cholera case admission is not always laboratory confirmed. Other research from DRC has shown that only a minority of those admitted to CTCs actually had cholera [57] and therefore this may skew some of our research findings in relation to experiences of the disease. As noted, cholera was often used by research participants to be a catch all term for diarrhoeal diseases, this emic construction may have therefore also distorted the way people described their experiences and perceptions in relation to the disease.

This research was conducted in partnership with Action Contre la Faim and for security reasons our research team were required to wear a branded vest throughout data collection and travelled in a branded vehicle. Given that the organisation have a history of working on WASH projects in this region and that participants had been exposed to decades of humanitarian response programmes, this may have increased willingness to participate and resulted in more socially desirable answers. The research team tried to reflect on this during daily research discussions and consider how our individual and collective positionalities may have shaped our findings.

## Conclusion

In addition to having severe health implications, outbreaks have the potential to disrupt people's social, psychological, and economic lives. By focusing on the lived experiences of cholera, our research highlighted that even when substantial shifts in behavioural determinants occur, it is not always enough to substantially influence the uptake of preventive behaviours like handwashing with soap. In this case, handwashing behaviour remained low during the outbreak due to the absence of enabling physical and social environments and the competing priorities and vulnerabilities of the population. Handwashing programmes targeting areas with endemic cholera or outbreaks within complex crises could be strengthened by acknowledging the underlying circumstances that create and perpetuate outbreaks, addressing the health and non-health impacts of diseases like cholera, investing in sustainable handwashing infrastructure, and identifying and sharing local disease coping mechanisms that facilitate the practice of preventative behaviours.

## Supporting information

**S1 Table. Handwashing determinant definitions adapted from on the BCD checklist of determinants (1, 2) and accompanied by method selections.**
(DOCX)

**S2 Table. Description and sample size for all methods done at a household or individual level.**
(DOCX)

**S3 Table. Purpose, description and sample size for each of the methods done within group discussions.**
(DOCX)

**S4 Table. Inclusivity questionnaire.**
(DOCX)

**S5 Table. Identified determinants and their associated influence on handwashing behaviour in Eastern DRC.**
(DOCX)

## Acknowledgments

We would like to thank the following people for helping to facilitate the research and contributing to ongoing reflections about emergent insights: Isiaka Hemedi, Justine Badhera Habamungu, Léon Ngwasi, Batian Arthur Benao, Marie-Paul Chirimwami, Karine Le Roch, Jean Lapegue and representatives of the Goma WASH Cluster. We would particularly like to thank the individuals who gave of their time to participate in this research and who welcomed us into their homes and shared their personal experiences so openly.

This research was undertaken as part of the Wash'Em Project which aims to improve handwashing promotion in humanitarian crises. The contents are the responsibility of the authors of the paper and do not necessarily reflect the views of our donors, USAID or the United States Government.

## Author Contributions

**Conceptualization:** Sian White.

**Data curation:** Sian White.

**Formal analysis:** Sian White.

**Funding acquisition:** Sian White, Thomas Heath.

**Investigation:** Sian White, Anna C. Mutula, Modeste M. Buroko, François K. Mazimwe.

**Methodology:** Sian White, Val Curtis.

**Project administration:** Thomas Heath.

**Supervision:** Karl Blanchet, Val Curtis, Robert Dreibelbis.

**Validation:** Anna C. Mutula, Modeste M. Buroko, Thomas Heath, François K. Mazimwe.

**Writing – original draft:** Sian White.

**Writing – review & editing:** Anna C. Mutula, Modeste M. Buroko, Thomas Heath, François K. Mazimwe, Karl Blanchet, Robert Dreibelbis.

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
