## [Decision Letter · Decision Letter 0]

2 Feb 2022

PONE-D-21-38155How does handwashing behaviour change in response to a cholera outbreak? A qualitative case study in the Democratic Republic of the Congo.PLOS ONE

Dear Dr. White,

Thank you for submitting your manuscript to PLOS ONE. After careful consideration, we feel that it has merit but does not fully meet PLOS ONE’s publication criteria as it currently stands. Therefore, we invite you to submit a revised version of the manuscript that addresses the points raised during the review process. The reviewers are generally impressed with the paper but have some specific comments for improvement which I hope you are able to follow carefully and respond to.

We look forward to receiving your revised manuscript.

Kind regards,

Alison Parker

Academic Editor

PLOS ONE

Journal Requirements:

4. Please amend your authorship list in your manuscript file to include author  Sian White, Anna C Mutula, Modeste M Buroko, Thomas Heath, François K Mazimwe, Karl Blanchet, Val Curtis, Robert Dreibelbis. 

5. We note you have included a table to which you do not refer in the text of your manuscript. Please ensure that you refer to Table 1 in your text; if accepted, production will need this reference to link the reader to the Table.

Reviewers' comments:

Reviewer's Responses to Questions

**Comments to the Author**

1. Is the manuscript technically sound, and do the data support the conclusions?

Reviewer #1: Yes

Reviewer #2: Yes

2. Has the statistical analysis been performed appropriately and rigorously? 

Reviewer #1: N/A

Reviewer #2: N/A

3. Have the authors made all data underlying the findings in their manuscript fully available?

Reviewer #1: No

Reviewer #2: Yes

4. Is the manuscript presented in an intelligible fashion and written in standard English?

Reviewer #1: Yes

Reviewer #2: Yes

5. Review Comments to the Author

Reviewer #1: Overall, this was well designed matched case-control study and the matching criteria were clearly laid out. However, there following would need some clarification.

Methods

1. Line 40-42 states the objective was to find out how the people responded to the cholera outbreak and how the outbreak affected their handwashing behavior. Is it right to assume that you wanted to find out if they washed their hands more often after the outbreak? How was this done/measured? was there a measurement of handwashing behavior before the outbreak to compare with that after the outbreak?

Results

From table 1, 21% lived in camps while the rest were lived in the community, was there a difference in water supply between the camps and community? Did this difference have any impact on the handwashing behavior?

Summary and general comments

Thank you for the privilege to review this well written paper. The authors carried out a matched case-control study to examine the hygienic behavior of the people of Kivu during the 2017 cholera outbreak. These results will be very useful for the cholera outbreak literature. I suggest revising for key elements in the methods and results which need a bit more clarification. Again I commend the authors for clarity of thoughts which allowed me to only focus on the science.

Reviewer #2: Referee review for: "How does handwashing behavior change in response to a cholera outbreak? A qualitative case study in the Democratic Republic of Congo”

This study uses a mixed methods approach to examine perceptions and behaviors associated with handwashing with soap among households that had experienced cholera and households that had not experienced cholera during a cholera outbreak. The outbreak took place in 2017 in a resource poor community in the eastern DR Congo where endemic and epidemic cholera exists. Health education messages involving biomedical information about the health benefits of handwashing had been disseminated over time in the study community. Study results suggest the handwashing with soap was rare during the outbreak among both case and control households despite self-reports of handwashing. The findings highlight that competing demands related to food consumption and livelihoods constrained people living in both case and comparison households from practicing handwashing. The authors conclude that severe psychological, social, and economic impacts that occurred in case households increased barriers to handwashing. The authors underline that handwashing strategies in contexts with endemic and outbreaks of cholera should consider emic understandings of disease and promote sustainable strategies that take into account social and physical environmental factors, including access to handwashing materials, that influence handwashing behavior.

Comments

1. Little information is provided on the cholera outbreak. It would be helpful to include a description of the outbreak including how long it lasted and the number of cases and deaths.

2. The manuscript would benefit from a description of the messages provided to community members, including how messages were disseminated, the content of messages and how this changed during the epidemic, the frequency of dissemination, and the target audience. Were the messages focused on diarrhea prevention or were cholera-specific messages disseminated?

3. It is not clear how focus group discussion participants were selected. Also, a description of the composition of the focus groups is needed.

4. While the authors mention that some households participated in more than one method, there is no indication regarding the extent to which IDIs were carried out in households where structured observations took place. The methods section should provide a description indicating when more than one method was carried out in one household.

5. Little is said about the differences between the IDP and local populations and how their living environments and daily routines affected exposure to interventions, perceptions of cholera, and handwashing practices. It would be helpful to include a description of the different social and physical environments, as well as exposure to messaging, to understand how this may have improved or constrained handwashing. Also, a description of differences in the way the outbreak impacted IDPs and the local population would be helpful.

6. The study results suggest the cholera is a general term used to describe a range of diarrheal illness. However, there is no mention of how this may have influenced emic views and the cholera explanatory disease model related to disease severity, causal explanations, and transmission, as well as the perceived need to wash hands. In other words, to what extent were informants able to distinguish disease signs, symptoms, severity, causation, and transmission from other types of diarrheal illness.

7. The authors rightly advocate for incorporating emic views and local constructions of disease into messaging and intervention strategies. However, the manuscript lacks some key information on the local explanatory disease models that could potentially be useful when informing strategies. I wonder whether some of this information was collected during the participatory methods that were not presented in the results. If so, it would be useful to include more details on the local explanatory model of cholera and how this may have affected handwashing behaviors. In that regard, what were the results of the participatory methods and how were the findings used.

8. In the discussion, the authors cite literature suggesting that behaviors may change over the course of an outbreak. It would be helpful to know whether this research detected any behavioral changes over time as perceptions related to the disease and risk of exposure may have changed.

Other comments

Recent research carried out in an IDP camp in North Kivu, which highlights similar findings related to the prioritization of competing needs over handwashing, should be mentioned.

Some statements made in the first paragraph of the discussion section do not necessarily reflect the study findings. Examples include:

- Cholera is conceptualized as a health challenge that is easily treatable

- Handwashing was hampered due to the fact that the work took place in a conflict zone

- Health promotion has led to high levels of knowledge related to the health impacts of cholera

6. PLOS authors have the option to publish the peer review history of their article (what does this mean?). If published, this will include your full peer review and any attached files.

Reviewer #1: **Yes: **Afoumbom Mildred Tita

Reviewer #2: No

---

## [Author Response · Author response to Decision Letter 0]

25 Feb 2022

We thank both reviewers for their encouraging words of support and their constructive ideas for strengthening this manuscript. We have explained how we have taken on their feedback below: 

Reviewer 1 Comments: 

7. Methods: Line 40-42 states the objective was to find out how the people responded to the cholera outbreak and how the outbreak affected their handwashing behavior. Is it right to assume that you wanted to find out if they washed their hands more often after the outbreak? How was this done/measured? was there a measurement of handwashing behavior before the outbreak to compare with that after the outbreak?

Response: No this is an incorrect assumption. The objective is broader than what reviewer 1 assumes. Given that this was an exploratory study we did not start with a priori assumptions about how behaviour may change during an outbreak. The study focuses primarily on the determinants of behaviour (rather than behaviour itself) and how these may have been shifted by the outbreak. While observational measures of handwashing are included, they were only on a small sample and were designed to be qualitatively analysed rather than to indicate handwashing prevalence or frequency. Similarly participants self-reported their behaviour during many of the methods but we know that self-reported handwashing behaviour tends to overestimate practice. We have not adjusted the objective but rather have added some text in the limitations to better explain this. 

8. Results: From table 1, 21% lived in camps while the rest were lived in the community, was there a difference in water supply between the camps and community? Did this difference have any impact on the handwashing behavior?

Response: This information has been added to the text. In summary there were no substantial differences. 

9: Summary and general comments: Thank you for the privilege to review this well written paper. The authors carried out a matched case-control study to examine the hygienic behavior of the people of Kivu during the 2017 cholera outbreak. These results will be very useful for the cholera outbreak literature. I suggest revising for key elements in the methods and results which need a bit more clarification. Again I commend the authors for clarity of thoughts which allowed me to only focus on the science.

Response: We thank Reviewer 1 for their kind words of encouragement. 

Reviewer 2 Comments

10. Little information is provided on the cholera outbreak. It would be helpful to include a description of the outbreak including how long it lasted and the number of cases and deaths. 

Response: While in DRC data on cases and deaths were shared with us by individual health centers, humanitarian organizations and the district government. However this data was inconsistent and hard to verify. Total case numbers and mortality from the national outbreak are available and have been included in the text but unfortunately due to these data issues we have not been able to include anything more specific for the region. ‘Outbreaks’ of cholera in DRC are, sadly unending, with this region reporting endemic cases every week of the year. Therefore and outbreak within the DRC context refers to a dramatic rise in cases as is typically experienced most years between about August and January. We have not given this detail in the text as we have already mentioned that our study took place during the peak of the outbreak during October and November. 

11. The manuscript would benefit from a description of the messages provided to community members, including how messages were disseminated, the content of messages and how this changed during the epidemic, the frequency of dissemination, and the target audience. Were the messages focused on diarrhea prevention or were cholera-specific messages disseminated?

Response: This was not assessed formally as part of this study. We have provided some additional description of this based on what participants told us. However we did conduct some parallel research with humanitarians about their programming throughout the Eastern region of DRC. This work has been submitted separately to Conflict and Health. 

12. It is not clear how focus group discussion participants were selected. Also, a description of the composition of the focus groups is needed.

Response: Information on sampling is provided under the heading ‘sampling’ in the methods section. We have expanded on this information about the composition of the FGDs under this section of the methods. 

13. While the authors mention that some households participated in more than one method, there is no indication regarding the extent to which IDIs were carried out in households where structured observations took place. The methods section should provide a description indicating when more than one method was carried out in one household.

Response: We have added this information under the IDI section. 

14. Little is said about the differences between the IDP and local populations and how their living environments and daily routines affected exposure to interventions, perceptions of cholera, and handwashing practices. It would be helpful to include a description of the different social and physical environments, as well as exposure to messaging, to understand how this may have improved or constrained handwashing. Also, a description of differences in the way the outbreak impacted IDPs and the local population would be helpful. 

Response: As noted above we have added some information on the similarity of WASH circumstances between IDPs and host communities. In the text we had tried to describe the different housing of both populations. Since these camps were informal and located within the heart of the communities there was little difference in exposure to hygiene promotion, however those outside the center of town were slightly less exposed as is now noted in the text. We have also included more detail on employment. 

15. The study results suggest the cholera is a general term used to describe a range of diarrheal illness. However, there is no mention of how this may have influenced emic views and the cholera explanatory disease model related to disease severity, causal explanations, and transmission, as well as the perceived need to wash hands. In other words, to what extent were informants able to distinguish disease signs, symptoms, severity, causation, and transmission from other types of diarrheal illness. 

Response: We have added content to the limitations section of the paper to address this point. We agree that this may have biased or distorted their reporting of experiences and perceptions towards the disease, however we don’t feel that this would have had a substantial bearing on understandings of disease causation and transmission given that this would be similar to many other diarrheal diseases (at least at a lay population level). Under the knowledge heading in the text I have added that while people viewed it as a catch all term people were able to easily identify cholera-specific symptoms such as rice-water stools. 

16. The authors rightly advocate for incorporating emic views and local constructions of disease into messaging and intervention strategies. However, the manuscript lacks some key information on the local explanatory disease models that could potentially be useful when informing strategies. I wonder whether some of this information was collected during the participatory methods that were not presented in the results. If so, it would be useful to include more details on the local explanatory model of cholera and how this may have affected handwashing behaviors. In that regard, what were the results of the participatory methods and how were the findings used. 

Response: As Reviewer 2 suspected the research aims and methods did not specifically seek to comprehensively understand local explanatory disease models. However quite a lot was learned about this during some of the participatory methods used in the IDIs and FGDs (e.g. personal histories, risk scaling, and cholera case descriptions). Most of these findings we have tried to capture in our results already. Specifically the following points are linked to this: 

- Cholera is seen as a challenge closely associated with poverty and hunger

- Cholera is associated with people who are thin, pale and sickly (ie these individuals are considered more vulnerable).

- Understandings of cholera were aligned with western biomedical models of understanding disease (this point has been added in text) given the history of health promotion in the region. 

- A minority of people felt that Congolese people are resilient to cholera or that if they got infected it wouldn’t cause them serious illness. 

- A minority of people felt that if you are too clean you would actually be more vulnerable to cholera. 

- A minority of people felt that it was in humanitarian and government interests for cholera outbreaks to continue.

- People who had had cholera tended to be more likely to see cholera as being random. 

In terms of the second part of your question, participatory methods were used in throughout each of the IDIs and FGDs (see supplementary materials for full descriptions). Therefore all the content included in the results relates to these methods. 

17. In the discussion, the authors cite literature suggesting that behaviors may change over the course of an outbreak. It would be helpful to know whether this research detected any behavioral changes over time as perceptions related to the disease and risk of exposure may have changed. 

Response: It is perhaps worth noting that the literature cited in relation to this point is limited by its quality. As noted in text that is because the majority of these studies rely on self-reported behaviour. As noted in the text the consensus across such texts is that behaviour seems to fluctuate over the course of many outbreaks and that one of the many factors driving this appears to be perceptions about the disease and the fear associated with it. This indeed is the premise behind Terror Management Theory which is referenced in the text. Given that we feel we have already explained this literature and its implications we have not added anything to this section. 

18. Recent research carried out in an IDP camp in North Kivu, which highlights similar findings related to the prioritization of competing needs over handwashing, should be mentioned. 

Response: We have added in some additional content into the discussion section based on the work of Blum et al 2019 and Claude et al 2020.

19. Some statements made in the first paragraph of the discussion section do not necessarily reflect the study findings. Examples include: a) Cholera is conceptualized as a health challenge that is easily treatable b) Handwashing was hampered due to the fact that the work took place in a conflict zone, c) Health promotion has led to high levels of knowledge related to the health impacts of cholera. 

Response: In relation to point a) this was mentioned in our results but has now been amended so this is a bit more explicit. In relation to point b) our discussion doesn’t quite claim that handwashing is hampered because of the conflict but rather the social and environmental factors that are associated with poverty, and the circumstances crated by conflict and displacement. We have tried to further clarify this nuance in the text. In relation to point c) We realize that while mentioned the high levels of knowledge in the results section we left out information on where participants felt this knowledge came from. This has now been added to the results section so this link is clearer.

---

## [Decision Letter · Decision Letter 1]

29 Mar 2022

How does handwashing behaviour change in response to a cholera outbreak? A qualitative case study in the Democratic Republic of the Congo.

PONE-D-21-38155R1

Dear Dr. White,

We’re pleased to inform you that your manuscript has been judged scientifically suitable for publication and will be formally accepted for publication once it meets all outstanding technical requirements.

Kind regards,

Alison Parker

Academic Editor

PLOS ONE

Additional Editor Comments (optional):

Reviewers' comments:

Reviewer's Responses to Questions

**Comments to the Author**

1. If the authors have adequately addressed your comments raised in a previous round of review and you feel that this manuscript is now acceptable for publication, you may indicate that here to bypass the “Comments to the Author” section, enter your conflict of interest statement in the “Confidential to Editor” section, and submit your "Accept" recommendation.

Reviewer #1: All comments have been addressed

2. Is the manuscript technically sound, and do the data support the conclusions?

Reviewer #1: (No Response)

3. Has the statistical analysis been performed appropriately and rigorously? 

Reviewer #1: N/A

4. Have the authors made all data underlying the findings in their manuscript fully available?

Reviewer #1: (No Response)

5. Is the manuscript presented in an intelligible fashion and written in standard English?

Reviewer #1: (No Response)

6. Review Comments to the Author

Reviewer #1: (No Response)

7. PLOS authors have the option to publish the peer review history of their article (what does this mean?). If published, this will include your full peer review and any attached files.

Reviewer #1: **Yes: **Afoumbom Mildred Tita

---

## [Editor Report · Acceptance letter]

1 Apr 2022

PONE-D-21-38155R1 

How does handwashing behaviour change in response to a cholera outbreak? A qualitative case study in the Democratic Republic of the Congo. 

Dear Dr. White:

I'm pleased to inform you that your manuscript has been deemed suitable for publication in PLOS ONE. Congratulations! Your manuscript is now with our production department. 

Kind regards, 

on behalf of

Dr. Alison Parker 

Academic Editor

PLOS ONE